# Physical-Literacy-Enriched Physical Education: A Capabilities Perspective

**DOI:** 10.3390/children10091503

**Published:** 2023-09-04

**Authors:** Elizabeth Durden-Myers, Gillian Bartle

**Affiliations:** 1The School of Education, Bath Spa University, Bath BA2 9BN, UK; 2The School of Education and Applied Sciences, The University of Gloucestershire, Cheltenham GL50 2RH, UK; 3The School of Humanities, Social Sciences and Law, The University of Dundee, Dundee DD1 9SY, UK; g.bartle@dundee.ac.uk; 4The Faculty of Social Science, The University of Stirling, Stirling FK9 4LA, UK

**Keywords:** physical literacy, physical education, education, capability approach, philosophy

## Abstract

(1) Background: Physical literacy is increasing in popularity across the world as a concept central to the promotion of lifelong engagement in physical activity across a multitude of sectors. The education sector has embraced physical literacy as a concept worthy of focus. Physical literacy literature is bold in its claim that physical literacy should be the foundation of physical education. The objective of this paper was to understand the value of physical literacy as the goal of physical education through the lens of the capability approach; (2) Positioning: This research adopted a post-qualitative sensibility whereby knowledge is decentered, favoring the inseparability of ethics, ontology, and knowledge (ethico-onto-epistemology); (3) Discussion: Throughout the discussion, traditional humanist examples are extended to include post-humanism perspectives to offer a more holistic and ecological appreciation of the relationship between capabilities, physical literacy, and physical education, using the ten capabilities of life, bodily health, bodily integrity, senses, imagination and thought, emotions, practical reason, affiliation, other species, play, and control over one’s environment; (4) Conclusions: The paper concludes with the recommendation that the capabilities approach offers a valuable framework for the continued justification of physical-literacy-enriched physical education, which, when aligned, can help to shape the opportunities provided for children and young people in support of their holistic development and lifelong engagement in physical activity.

## 1. Introduction

The term physical literacy (PL) has been used as a vehicle to raise awareness of the importance of physical activity within societies and populations that are becoming increasingly sedentary over the world [1]. PL is a multi-sectoral concept that transcends health, education, sport, and leisure [2], among others. It is also positioned within education as the “fundamental goal of physical education” [3] (p. 162) [4]. Corbin [5] (p. 24) also states that: 

“Those interested in promoting physical literacy will have to show that the adoption of the term physical literacy provides a foundation for elite sport, public health, and physical education rather than merely being a term used to improve public perceptions”.

However, perspectives regarding the rationale for PL to be considered the foundation of physical education (PE) primarily focus on the avoidance of hypokinetic diseases and maintaining the body as a machine or object [3]. Although important, this message fails to acknowledge the living body [6] and the full conceptualization of PL as a holistic concept that enhances both being well (health) and living well (flourishing). To articulate the full value of PL, it is important that both the body as lived and the living body are acknowledged. Whitehead [3] and Whitehead, Durden-Myers, and Pot [7] discuss the value of fostering PL, drawing from the perspectives of philosophy, neuroscience, social justice, human development, psychology, and sociocultural fields of study. In both articles [3,7], PL is positioned as a human capability, the “development of which can be viewed as a human right” [7] (p. 254).

The capability approach is also a perspective used to guide the benefits derived from education [8]. Bessant [9] (p. 145) argues that applying the “Capability framework entails valuing education for its intrinsic goods rather than for its instrumental importance”. A capability perspective highlights the role of freedom and choice in bringing about the “kind of life we value and the capabilities we need to enjoy for a life we treasure” [10] (p. 227). It would mean assessing whether education affords the capabilities needed to achieve a full and flourishing life rather than the mere obtaining of materialistic or detached goods (that is, grade levels, estimated income, further education, or employment statistics) [9].

Both PL and the capabilities approach offer complementary perspectives that challenge the dominant utilitarian (outcome-orientated) approach to education and PE. In this paper, we look to rationalise the value of physical literacy-enriched physical education (PL-enriched PE) through the perspective of the capability approach [10]. We address Robeyns’ point [11] (p. 40) that: 

“Relatively little work has been done on the question of the conceptual properties of capabilities understood as freedoms or opportunities and on the question of the minimum requirements of the opportunity set that make up these various capabilities”.

Regarding Robeyns’ call, the freedoms or opportunities offered by PL-enriched PE are not solely determined by individual factors. Rather, a multitude of human and nonhuman elements contribute to PL-enriched PE [12] and the development of our capabilities. We seek to extend the relationship between capabilities and PL with examples from both humanist and posthumanist perspectives relevant to PE. Moreover, by placing PL as a central human tenet as a focal point or fulcrum through which many other literacies pass [13], we acknowledge this as an anthropocentric humanist position. However, this is merely to clarify the relevance of PL for human capabilities that begin at birth, in moving [13,14]. We detail our posthuman approach to PL below. First, we discuss the capabilities approach as a different way of valuing PE as educationally worthwhile.

What constitutes an educationally valuable subject or activity may be positioned as any activity that develops human potential, capability, or capital required to lead and live a fulfilling and flourishing life [15]. Central to this statement is the idea that human capabilities are those that are required to live a fulfilling life. It seems that the nurturing of these capabilities could therefore be considered a fundamental human right [16]. If PL could be considered a capability, or significant in the nurturing of existing defined capabilities, it could be argued as having educational value. Durden-Myers [4] (p. 52) states that: 

“the development of physical literacy within physical education is a valuable endeavor in its own right, and that physical education has the capacity to develop our embodied capability, which should be an integral part of a holistic education”.

Before we tackle this debate, we first acknowledge ourselves as being entangled with this research [17] by outlining our positioning within this paper.

## 2. Positioning

Elucidating how/whether a capabilities approach relates to PL-enriched PE, we adopted a post-qualitative sensibility [18]. In eschewing a predetermined methodology, or refusing methodology, we sought to achieve three things. First, our approach is primarily ontological to avoid the association of knowledge as purely an epistemic exercise [19,20]. Second, we employed philosophical concepts to interrupt thinking [21] about PL and how it might be discussed alongside understandings of PE. We are thinking with theory [22]; in this case, with the narrative of the capabilities approach as well as previous research literature. Third, ‘thinking-doings’ in PE and PL is the pedagogical methodology that we think is at work whilst PE and PL are enacting—in other words, practice and theory are interwoven, inseparable, and emergent [23]. Importantly, concepts are dynamic and not fixed but rather come to be known as they are becoming visible [20].

By embracing both humanist and posthumanist perspectives, we extend and balance the conceptualisation of capabilities beyond the limitations of traditional approaches. Although Nussbaum [16] includes nonhuman animals in her capability approach, it retains a humanist, anthropocentric perspective. In our context too, humanist perspectives emphasise the importance of individual agency and self-determination in shaping PL. Meanwhile, posthumanist perspectives highlight the role and agency of technology and other nonhuman factors as ‘entangled’ with human (cap)abilities [24]. By incorporating both perspectives, we can retain the holistic understanding of PL but extend this to consider relevant things, whether non/human and/or inanimate. Ultimately, this approach allows us to broaden our conception of what it means to be physically literate and is a different way of conceiving how human and nonhuman perform capabilities. Phenomenological concerns with human experiences are extended by implementing Deleuze’s idea of becoming [19,25], wherein connections or relations among things human and nonhuman bring different significance to PL [12]. In the current paper, we include nonhumans and do so by attending to circulations of relations to consider whether they are enacting PL-enriched PE and capabilities or not.

Finally, there is a fundamentally ethical concern wherein following a capabilities approach, such as offered by Nussbaum [16], is in itself a virtuous practice. Barad’s [24] terminology might help in that we adopt an ‘onto-ethico-epistemological’ sensing when discussing the potential worth of a capabilities approach and PL-enriched PE. The onto-ethico-epistemological awareness also captures our positioning as researchers already entangled with the world [17]. This paper is designed to provoke thought among professionals and representatives within education and other institutions, concerning PL and the purpose and value of PE. It is also hoped that following a capabilities approach provides another way to access understanding of PL or to articulate more clearly the value of PL for education and, more specifically, as the ultimate aim of PE.

## 3. The Capability Approach, Physical Literacy, and Physical Education

The capability approach is closely linked with the human rights approach [16]. The relationship between the capability approach and the human rights approach “lies in the idea that all people have some core entitlements just by the virtue of their humanity” [16] (p. 62). The capability approach (or sometimes capabilities approach [11]) can be viewed as a means to integrate human rights approaches and serve as a medium to identify human entitlements and therefore responsibility. Nussbaum [16] (p. 41) highlights that: 

“(T)he moral relevance lies in whether capabilities are truly available to us given the choices made by others, since that is the real freedom to live our lives in various ways, as it is truly open to us”.

By virtue of the idea of an entitlement comes with it a responsibility to provide opportunities for the fulfilment of that entitlement. However, assigning responsibilities to specific people or groups can be problematic on several levels. For example, for an entitlement to be made available to all it requires a shared philosophy by governmental and political structures. Here, we would argue for the inclusion of all living things and how these are always complicit with, or in sympathy with [26] nonhuman things for life to flourish.

A further concern is that some uses of the term ‘entitlement’ are based on an anthropocentric—and androcentric [26]—notion of being in the world, arguably at the expense of a posthumanist perspective of being of the world. Although the capabilities approach does advocate for environmental care and respect [16], it does so from a humanist perspective with some acknowledgement of the importance of other animal species. Posthumanist understandings of capabilities might foreground other features of living and being well with the more-than-human world, including inanimate things [27]. With this last point in mind, we aim to explore the current assemblages of PL in relation to the capability approach and offer new lines of flight [25,28] throughout this paper, entangling examples from humanist and posthumanist perspectives.

The capability approach can be described as a way of comparatively assessing the quality of life and of theorising about basic social justice [16]. In other words, the approach takes each person as an end, asking not just about the total or average wellbeing but about the opportunities available and moreover what opportunities should be made available to promote or form the foundations from which quality of life may be achieved [16]. This is very similar to the concept of flourishing in that it is not purely focused on the ends (quality of life or flourishing) but also with understanding and valuing the means to achieving this end, and what opportunities might be provided. The capability approach may serve as a lens to promote conditions conducive to flourishing, and to understand characteristics of those who consider themselves to be flourishing. Furthermore, Nussbaum [16] asserts that developing a defined range of capabilities (outlined below) is concerned with ensuring quality of life. Similarly, Robeyns [11] (p. 9) suggests that the approach asks “what people can do and be (their capabilities) and what they are actually achieving in terms of beings and doing (their functionings)”. It seems that certain features are present for flourishing, rather than merely striving for an ideal of this. Nussbaum [16] goes further, suggesting that, in the absence of any capability, the individual will not have achieved a fully human existence. Therefore, capabilities must be given the opportunity to develop, since without this humans will fail to experience quality of life and will not achieve a fully human existence. If this is the case, then, to flourish, these capabilities ought to be developed.

Nussbaum [16,29] and Sen [10] identify ‘beings and doings’ as avenues of interaction. Nussbaum refers to these as capabilities, while Sen prefers to talk about ‘functionings’. Together, they argue that every individual should have the right to realise these human potentials, and, not only should they be provided with the opportunity to exercise these capabilities, but they should also have the freedom to do so if they so wish. Nussbaum [29] (p. 97) describes that “capabilities as I conceive them have a very close relationship to human rights”. Nussbaum [16,29] and Sen [10] put forward similar calls to action from the government to facilitate the development of capabilities, seeing them as fundamental human rights. Nussbaum specifically identifies a range of capabilities, which include language, each of the arts, numbers, scientific aspects of the world, interacting with other people, and historical understanding. She includes the embodied dimension within very many of her capabilities. However, she presents her thoughts from a dualist perspective and casts the embodiment, in most cases, as facilitating the realisation of other capabilities.

It is argued here that, as a key medium to interact with the world, the human-embodied dimension as both the lived body and the living body, and as endowed with perceptuomotor abilities, has a strong case to be acknowledged as a capability in its own right [7]. As Robeyns [11] and Sen [30] also note, capabilities are the actual freedoms that people enjoy as part of how they choose to live their lives, without needing another resource to actualise that freedom. The embodied dimension affords us a unique, fundamental, and highly significant mode of interacting with the world, and, in line with other capabilities, every human has the right to draw upon and be afforded opportunities to develop this potential [4]. As capabilities are developed and drawn upon throughout life, it enables the individual to blossom and thrive [16,29,30]; therefore, it would seem to follow that opportunities should be fostered in schooling to promote capabilities. If the relationship between nurturing PL and the development of capabilities can be made, then this may provide support for advocating the educational value of PL, thus creating a platform to argue for the value of promoting physical activity throughout life within PE.

PL has often been referred to as a capability [3] and a potential that all human beings possess.

“Physical literacy supports the view that we should celebrate our embodied capability, a capability that needs no justification beyond its unique and indispensable contribution to human life”[31] (p. 27).

PL when described as a capability articulates the expression of our embodied dimension as one aspect of our innate human nature [3]. As human beings, we comprise a range of interrelated dimensions, such as physical, sociological, emotional, and psychological, to name but a few, through which we interact with the world [3]. The deployment of these dimensions can be considered as a particular capability. Nussbaum [16] categorises the basic capabilities as life, bodily health, bodily integrity, senses, imagination and thought, emotions, practical reason, affiliation, other species, play, and control over one’s environment, and states that all these capabilities are concerned with ensuring quality of life. PL or, more specifically, our embodied capability, has a direct connection with each of the ten capabilities outlined later in this article. Next, we explore the relevance of a capabilities approach for the context of PE.

Robeyns [11] highlights how in recent decades the capability approach has also been used as a framework for wellbeing, the freedom to achieve wellbeing, and the public values in which either of these can play a role, such as development and social justice. Robeyns [11] (p. 40) explains further, saying that “little work has been done on the question of the conceptual properties of capabilities understood as freedoms or opportunities and on the question of the minimum requirements of the opportunity set that make up these various capabilities”. PE has a clear role to play here in providing some minimum entitlements for all children and young people in support of their holistic development.

“Physical education is the name of a designated area of learning within the curriculum and is the ‘name’ of a subject area. Physical literacy, on the other hand, identifies an overarching concept promoting engagement in physical activity for life. Physical education, therefore, offers a unique opportunity for learners to have educational experiences that may develop their physical literacy”[4] (p. 47).

PE, when enriched by PL, can help to provide a more socially just PE ecosystem by concentrating on a range of positive and meaningful experiences, whereby the individual, inclusion, and holistic development are valued and nurtured [32,33]. Alignment between PE and PL would therefore also provide some minimum entitlements and opportunities for the development of capabilities. This relationship is explored in the following section.

## 4. Exploring the Relationship between Capabilities, Physical Literacy and PE

As mentioned earlier, Nussbaum [16] (pp. 33–34) categorises ten ‘central’ capabilities as: (1) life, (2) bodily health, (3) bodily integrity, (4) senses, imagination, and thought, (5) emotions, (6) practical reason, (7) affiliation, (8) other species, (9) play, and (10) control over one’s environment. Nussbaum [16] argues that all of these capabilities are concerned with ensuring quality of life. Each of these capabilities are discussed in the following paragraphs, including the contribution of PL to each capability and how this may be developed through PL-enriched PE. As can be seen thus far, much of the capability approach rationale, positioning, and examples proposed by Nussbaum [16,29] and Sen [10] are from a humanist perspective whereby the human is positioned centrally in the narrative. Throughout the discussion below, exemplifications are considered from both humanist and posthumanist perspectives.

## 5. Life

Life as a “capability is concerned with being able to live to the end of a human life of normal length, this means that the natural life line is not hindered through premature death or that life itself is not considered worth living” [16] (pp. 33–34). Factors that cause premature death include hypokinetic diseases and conditions that occur from a sedentary lifestyle or lack of physical activity. Examples of such diseases could include obesity, cardiovascular disease, and other complications arising from sedentary behaviour and physical inactivity. To help ensure that life is nurtured, education is one way to develop the knowledge, understanding, and skills necessary to navigate through life, as a part of how one might avoid adopting behaviours that may hinder one’s life [16].

Life as a capability can be related to the objectively good characteristic of human flourishing in that the activities people pursue in the desire to achieve flourishing or to develop a particular capability should not stunt or detrimentally affect ‘the natural life’ [16]. For example, occasionally consuming alcohol or processed food may be interpreted as an objectively good activity perhaps due to the socialization benefits or as a treat. However, regularly consuming many units of alcohol to the point of inebriation or regularly consuming processed food would be considered detrimental to ‘the natural life’. There may be negative effects on natural life in the long term, and this could not be considered objectively good. This example could also be used in relation to bodily health.

Life as a capability is also concerned with the potential to live in harmony with the world and with things nonhuman in the world [34]. The posthuman perspective also highlights the interconnectivity between capabilities as harmony with the world and as affiliation with/in/of the world and environment. It also speaks to a more controlled or respectful approach with the environment in the sense that the world provides the conditions for life to exist [26]. In this last sense, human-centric notions are replaced with posthuman, eco-philosophical ways of considering interactions in the context of PL-enriched PE.

Life is unequivocally connected to PL and the development of our embodiment as the very means through which we experience life, as well as being an integral aspect of our human nature that is required to live a natural life of normal length. Without an embodied sense of self, sustained involvement in physical activity is unlikely [3]. Lifestyle choices that are conducive to living a worthwhile life of normal length might not be achievable if things in the environment and political ideology, for example, do not foster participation in physical activity throughout life [11]. PE may be used as the medium to promote this educational endeavour, as it has been described as a legitimate educational practice that has the capacity to influence young people’s lifelong engagement in physical activity [35].

## 6. Bodily Health

Bodily health is considered as the capability to be able to have good health, including aspects such as having access to adequate shelter and to be adequately nourished [16]. This capability is concerned with tackling issues affecting health; for example, combating poor nutrition and malnutrition, and educating societies about healthy eating. It seems that this is increasingly important as the pressures of some western lifestyles encourage the consumption of convenient and affordable food that is often processed and calorie dense.

PL and PE can develop bodily health through promoting a sustained involvement in physical activity throughout life and by developing the knowledge and understanding regarding fitness and health, including exercise, nutrition, and sleep [3]. Undertaking physical activity within PE lessons and learning about the benefits and value of being physically active can both help in the promotion of physical health. In fact, multiple countries worldwide are linking PE increasingly to health as a curriculum area or topic. Recent examples include the Australian (National) Curriculum (Australian Curriculum, Assessment and Reporting Authority) [36] and Canadian State of Ontario [37]. In Scotland and Wales, PE is located within the curriculum area of Health and Wellbeing [38,39]. While PE certainly has a role in the promotion of health and wellbeing, Kirk [40] warns of aligning too closely to this theme as PE may become accountable for the wider health agenda. The significance and value of PE could potentially be destabilised if it is not able to ‘make good’ on wider health claims.

Bodily health is, of course, not solely relevant to the physical dimension, and a posthuman approach brings to the surface the highly complex and interconnected political aspect of health and wellbeing. To avoid a dualist separation of human–nonhuman/nature–culture, it is worth being reminded of PL-enriched PE in the monist and holistic sense intended by Whitehead [3] and inviting to the discussion the work of posthumanist thinkers such as [26]. Bodily health is multifaceted and embraces flows of relations among cognitive, affective, physical domains, and nonhuman things. It is a highly political capability, which informs the discussion referred to above about whether wider health (and other) agendas can be included or not in the school subject of PE. It might be impractical to include debates about how political forces from around the world (democratic, totalitarian, or other) are affecting bodily health in the limited and already crowded curriculum area of health and wellbeing and, specifically, PE. The posthuman perspective on the importance of relations among all bodies (human and nonhuman) is also relevant for the capability of bodily integrity.

## 7. Bodily Integrity

Bodily integrity is concerned with the ability to move freely from place to place, without fear of violence, and being able to have the opportunities of satisfaction and choice in matters of sex and reproduction [16]. This may go further in terms of the ability to express oneself freely as long as it is moral and virtuous without fear of judgement. This aspect brings elements of social justice to the foreground, ensuring that people have the opportunity and right to live in a free and just world without persecution or oppression.

PL promotes bodily integrity through the confident expression of movement in a range of environments [3] that also engenders an embodied self-expression that communicates our thoughts, feelings, and emotions through embodied interactions. It values as one of its core principles the notional inclusion and appreciation of everyone’s unique PL journey [3]. This core principle is evident in the capabilities approach as discussed by Robeyns [11], where she addresses how the approach accounts for diversity for individuals to achieve wellbeing.

This is particularly important for PE because elements of society are often ‘played out’ in PE, where social injustice may thrive; for example, in the exclusion of or lack of provision for Special Educational Needs and Disability (SEND) pupils and in the lack of (equal) access to opportunities of choice [8]. The PE environment can be a stronghold for the reinforcement of societal and gendered norms, thereby restricting some individuals’ unique PL journeys. Experiences in PE may be fraught with opportunities for ‘othering’, in the sense or act of treating someone as though they are not part of a group and are different in some way. The point here is that equal opportunities still fall short of equity unless the environment where opportunities are provided are equitable, safe (physically and emotionally), welcoming, inclusive, and socially just.

At individual, community, and societal levels, a capabilities approach seeks to change collective experiences of those whose bodily integrity might be marginalised; for example, women’s and indigenous groups, where intent to strive for bodily integrity will be different across societies and cultures. A posthuman perspective would investigate how nonhuman relations or forces are also factors in how bodies are silenced or lauded. In this sense, it becomes clearer that humans are not separate from other bodies, that they are not exceptional [41], and that bodily integrity takes on a different meaning from the humanist capability point of view.

## 8. Senses, Imagination, and Thought

Senses, imagination, and thought are concerned with being able to use the senses, to imagine, think, and reason, and to make sense of the world [16]. These are informed and developed through adequate and holistic education systems and experiences. Examples include being able to have imagination in thought and using this in connection with experiencing and producing works and events of one’s own choice; further, being able to have the freedom to practise one’s religion and beliefs and to practise free speech and expression, artistic and political, whilst still demonstrating an awareness and respect for this freedom in others. Finally, Nussbaum [16] (p. 33) adds “Being able to have pleasurable experiences and to avoid nonbeneficial pain”.

The use of and development of our senses, imagination, and thought are all developed within PL by developing an embodied sense of self. The development of perception and observation of the world around oneself is achieved using all senses as available to each individual. Imagination and thought are developed through the trial and error of creativity, intelligent action, and planning and overcoming problems.

Opportunities for exploring the range of sensory devices afforded to humans from birth are fundamental in PE. The subject area offers a range of environments, equipment, and relationships with others to play and practise physical activities safely and in a supportive ethos. Structured PE can provide opportunities not accessible to some children and young people, thereby extending the family or community offerings or overcoming limitations. This concurs with the contentions that capabilities approaches have with social justice by centralising education as a way of challenging limitations that people may be facing. Again, the importance of nonhuman things in the PE context is made visible; for example, if pupils learn to send a ball to a target, where they might not have had this relationship in their home lives. PL-enriched PE exposes children and young people to physical sensorial experiences as well as how they might come to know and understand through touch, sound, smelling, seeing, and doing, but this also garners broader affective development.

## 9. Emotions

Emotions are concerned with “being able to have attachments to things and people outside ourselves”, to love, to grieve, and to experience longing, gratitude, and justified anger or frustration [16] (p. 33). Emotional development or emotional intelligence is essential if we are to live a life that is full and avoid a life that is blighted by anxiety, fear, depression, or paranoia. This capability can be closely related with the agent relative and social construction characteristics of flourishing in that our emotions may cause us to form attachments with others but this will depend heavily on that individual as the agent, and their motives for seeking out, developing, or maintaining attachments. As well as the capability approach, human flourishing also places an importance on the interaction with others and considers this to be an integral part of our human nature [7]. This means that interactions with others (animate and inanimate) and our emotional response, and therefore attachment, will be entwined between the self and the environment [42].

PL and PE can provide opportunities for the exploration of emotions through physical activity and interaction with others. The relationship between emotions or ’affective states’ in relation to sustained engagement in physical activity has been arguably overlooked but, in the last 15 years, the interest of researchers in affective determinants of physical activity has increased [43]. Intuitively speaking, a deep and personal connection with movement and physical activity would be formed as a result of a bank of lived experiences associated with positive emotions. Wienke and Jekauc [44] examined the conditions under which enjoyment and positive emotions emerged during physical activity. They identified four factors—perceived competence, social interaction, novelty experience, and perceived physical exertion—that were associated with positive emotions. PL also highlights the importance of promoting motivation and confidence as determinants of lifetime engagement in physical activity [3]. These recommendations have real significance for the delivery of PE in moving beyond what activities are delivered in PE, and instead elevates how these activities are received by participants and the nature of the experiences themselves.

Much of Nussbaum’s [16] human-centric text relies on a social constructionist perspective. However, where she claims that attachments can be formed with ‘things and people’, posthumanism opens other ways of viewing capabilities. Social constructionism places mind above body and privileges discourse over enactment [45]. In PL-enriched PE, human–nonhuman are in perpetual motion when enacting PE environments, which can still, of course, promote a sense of belonging or alienation. To include all things—materials, the digital, the social, emotional, imagination, and sensorial—as already entangled in the environment acknowledges the complexity and messiness of PE. This posthuman perspective brings complexity to the surface and might help in considering different ways in which we can create a more welcoming and inclusive environment where more pupils feel they belong.

## 10. Practical Reason

Practical reason is the capability to conceive of and engage in what is considered a ‘good life’, to be able to reflect and alter one’s life according to one’s beliefs and freedoms. This capability is viewed, along with affiliation, as permeating all capabilities because, for example, people may be “well-nourished but not empowered to exercise practical reason and planning with regard to their health” [16] (p. 39). Therefore, people are not achieving human dignity since practical reason as a capability includes being able to reflect, choose to act, and have the freedom to enact choices. A posthuman perspective of this capability makes sense because humans are already entangled with their environments and will contemplate and reflect about planning lives being mindful of things in their environments. Posthumanism does not privilege humans as sole agents in these relationships [26]. For instance, thinking and planning for moving between places to grow food or find water necessarily involves human-with-nonhuman in practical reasoning about carrying out these activities.

Practical reason is the ability to form a conception of what can be considered good or virtuous and to engage in critical reflection about the planning of one’s life. Practical reason forms the cornerstone of the protection of liberty of conscience and religious observance [16]. In other words, higher-order thinking skills need to be developed in order to critically analyse, evaluate, reflect, and plan one’s own life.

Practical reason is central to PL since the latter refers to the capacity individuals have to choose to be active throughout their lives. Experiencing moving in a variety of environments develops the whole person, not only physical competence; that is, as people experience moving in a variety of environments, they are moving–thinking–feeling concurrently. Aristotle’s [46] concept of practical wisdom—phronesis—embodies what is meant here. Whilst becoming confident, developing knowledge and understanding about the value of physical activity, and having motivation to remain active throughout life, the capability of practical reason is enacting.

PL-enriched PE provides ample opportunities to explore movement experiences that are positive, fun, enjoyable, meaningful, and have the potential to shape how individuals continue to value being physically active [47,48]. Teachers and pupils could explore reasons for being well and how to adapt and plan for being healthy. Pedagogical approaches should be inclusive, thereby embracing differences, whether cultural, spiritual, or personal movement preferences and abilities. There is also a plethora of opportunities to explore, test, and adjust reasonings through movement within PE, whether adapting a set play to outwit an opponent, changing your grip in response to the rock face while climbing, altering your pacing during endurance running, or reflecting on how to enhance your gymnastic/dance routine. Practical reasoning could be closely linked with reflexivity; the ability to examine your own feelings, reactions, and motives (or reasons for acting) and how these influence how you think, how you respond, or what you choose to carry out in a given situation. PL-enriched PE encourages frequent opportunities for individuals to reflect on their PL journeys and think reflexively about their ongoing and future participation in physical activity.

## 11. Affiliation

Affiliation is separated into two areas but permeates all capabilities because dignity is its central premise for Nussbaum [16]. Firstly, being able to live with others and to recognise and show concern for other human beings, to engage in various forms of social interaction, and to be able to be empathetic. Secondly, affiliation is having the social basis of self-respect and being treated as a dignified human being whose worth is equal to that of others—in other words, being non-discriminatory on the basis of race, religion, ethnicity, sex, sexual orientation, caste, national origin, or age [16]. This capability draws attention to the humanistic values that the approach adopts, valuing each human equally and again drawing attention to the socially constructed world in which one lives.

The affiliation capability is also extended to a capability that is responsible for the care of other species. This capability is described as being able to live with concern for and in relation to animals, plants, and the world of nature, including natural resources and the world’s ecosystems [16]. There appears, then, to be a logical further extension that includes non-living things, thereby thinking of affiliation as permeating the inanimate as well as animate world.

Affiliation can be promoted when developing PL by providing opportunities for interaction with others and through nurturing empathic interaction with others; this comes as a direct result of a developing sense of embodiment. PL is also inclusive and looks to develop every individual regardless of their age, ability, gender, race, and so on, and aims to develop the movement potential within all. Through the exploration of a range of environments, PL can also engender positive relationships with other species. By walking a dog, for example, the result is a positive shared experience. Thoughts and feelings of positivity may be mutually experienced that, over time, may lead to further appreciation towards other species.

Through PE, outdoor learning, nature-based learning, or adventure activities form part of pupils’ experiences in formal [49] and informal school curricula around the world [50]. These experiences are intended to engender enjoyment within the natural environment but also include physical activities that rely on mutual concern, often for safety; for instance, pupils supporting one another through holding onto a rope when learning to traverse rocky terrain on a mountain walk. Without the environment (and the inanimate rope and supportive footwear) the activity may not be possible and therefore the experience would not happen. A variety of these experiences can lead to people who enjoy being physically active outdoors appreciating not only human–human affiliation but also learning why and how to protect the environment. Concerns such as these feature in posthuman and materialist accounts of education across the literature (for example, [51]). The importance of relations that are working to either destruct or re-construct natural resources is well documented. It seems logical to make the case that the capabilities approach supports dignity, care, and wellbeing for human and nonhuman things in the world.

## 12. Play

Play is described as being able to laugh, to play, and to enjoy recreational activities for therapeutic and development purposes [16].

As already mentioned above, as part of the contribution to human flourishing, PL is relevant from pre-birth [7], and the earliest interactions with the environment will be through sensory exploration and through play [14]. In the early stages, play is spontaneous and pre-reflective, as children interact with things in their immediate environments [52]. As conscious intentionality develops, children might choose or be encouraged to engage in physical activities [42]. PL and the capabilities approach overlap here, as each prompts engagement in movement and physical activity for its own sake. Nussbaum [29] relies on Aristotle’s [46] phronesis or practical wisdom/practical reasoning to show that playing, physical activity, and active choices throughout life are performative, ‘doings’, or tacit approaches to being of the world. PL literacy values the motivational, therapeutic, and developmental contribution that is possible through play.

During more structured PL-enriched PE, children are afforded opportunities to continue developing embodied experiences. For example, in the context of school PE, ‘play’ involves knowing and understanding rules of an activity whilst engaging in that activity. The inherent goods of choosing to play in an adventurous activity environment, to test oneself in a fitness context, or to contest a game contribute to enriching life [53,54] and to flourishing in the broadest sense. Hopefully, positive experiences during PL-enriched PE encourage individuals to keep being physically active; essentially, to keep playing.

From a capabilities perspective, play is a dynamic idea and an entitlement that has, at heart, the idea of fun, laughter, enjoyment, and freedom to choose. As environments change around a person, family, and community, so do the nonhumans with which we interact. Taguchi [52], for instance, draws attention to the materials that invoke playful activities, such as ‘sand-with-girl’, where the fistful of grainy soft stuff affords the sensory experience of letting it flow through the girl’s fingers. Recognising this posthuman perspective of play means allowing playful movements and physical activities to emerge not only with other humans but also with things in different environments. PL-enriched PE includes fostering freedom to play, encouraging an understanding of how to adapt at one’s own pace, and to realise the self with others and things in ways that provide fun, laughter, and enjoyment.

## 13. Control over One’s Environment

Finally, Nussbaum [16] considers control over one’s environment as a capability, separated into two categories: political and material. The first is being able to participate in political choices governing one’s life. The second is being able to hold property (both land and moveable goods) and have property rights on an equal basis with others. This capability also says that individuals ought to have equal capacity to seek employment, be able to exercise practical reason within the workplace, and to enter “meaningful relationships of mutual recognition” [16] (p. 34) with others. Control over one’s environment as a capability retains the principle of each person as an end in themselves. However, humans are both one part of and are also creating the political environments alongside inanimate things [24,41]. Thus, posthumanism problematises the idea of control over one’s environment since to rely on humans only seems to not recognise the relevance and contribution of things like structural devices or policy documents.

Control over one’s environment is promoted within PL through the appreciation of the agent-relative, existential, and phenomenological constructs, meaning that, when appropriate, individuals should be given the opportunity to make their own choices that govern their lives. Past experiences and interactions in movement contexts have already been shown to involve more than only human bodies, and so we reiterate here that decisions include social and material things. Decisions such as how people wish to be physically active by pursuing an activity of their own choice or ultimately choosing whether or not to be physically active are never taken alone. For instance, being hit on one’s body by a hard hockey ball might put one off playing that game—or any team game—again.

It is incumbent upon teachers of PE that movement experiences are meaningful, enjoyable, and positive for pupils as this increases the likelihood of individuals seeking to remain physically active [55]. PE has a place in many curricula but, in some cultures and countries worldwide, has been viewed as a low-status subject with only relative importance [56,57,58]. Further, children might not have the freedoms to influence socio-political contexts. The challenge for the PE community is to elevate the status and value of the subject from one with a relative value to a subject of an equal value, and one that is integral in providing a holistic education. Making the link between PE, PL, and developing capabilities may be one such avenue to achieve this. PL and PE does and can offer inclusive experiences with opportunities to interact with others in meaningful and respectful ways.

## 14. Discussion

One of the main tenets of the capabilities approach is to develop and encourage social justice through the identification of a range of capabilities that enable a flourishing life. Nussbaum’s [16] (p. 35) capability approach is individualist in that “the goal is to produce capabilities for each and every person, and not to use some people as a means to the capabilities of others”. Similarly, PL is initially an individual’s physical activity life journey. Both philosophies are also largely anthropocentric, although we fully recognise that Nussbaum [16] and Whitehead [6] make explicit the involvement of other animal species and the interplay between the self and the environment in their own works. The individualistic nature of these approaches is to acknowledge individual variation, heterogeneity [8]. As such, each approach is helpful for learners in PE since differences and choice can be incorporated into physical activity experiences. These ideas are significant for wider concerns such as social justice and equality.

There is merit in understanding the unifying constructs that provide the foundations for life to flourish. There is a need for ‘big’ theoretical ideas about social justice, fairness, and tools with which to evaluate educational activities [59]. The capabilities approach offers a perspective through which life may be evaluated [16]. PL-enriched PE as discussed above can provide many opportunities through which capabilities may be nurtured and thus unlock the potential for a full and flourishing life. Identifying the correlations, intersections, and relationships between these concepts is useful in understanding how the concepts are similar but also perhaps different, or indeed incomplete and ever-changing. We offer a critique here that the anthropocentric view is only one part of the full picture. Walker [59] (p. 168) suggests that “while social justice must be for individual flourishing, it should also be for collective solidarities, the one with the other”. We have therefore introduced posthuman perspectives throughout the discussion above to challenge the dominant individualist view of social justice within the capabilities approach. In this way, all bodies (not just human) become relevant because no human being is alone, and the relations with(in) the environment are emergent [45]. We position that, inter alia, the capability approach as a conceptual framework, alongside PL and PE, may be enriched by being extended to include the view that human and nonhuman life should flourish and embrace the relations among animate and inanimate as a connected life–world.

Previously, we mentioned how PL could be considered a fulcrum through which PL-enriched PE and capabilities interact. In Figure 1, the ‘see-saw’ metaphor depicts how balance between PL-enriched PE, PL, and capabilities can be achieved when PL acts as the fulcrum.

When balanced and aligned, there is a connectedness and flow between PE-PL and capabilities; for instance, the common goal of developing a love of and valuing movement for its intrinsic worth, increasing the possibility of sustained engagement for a flourishing life. There may also be instances of imbalance/disharmony; for example, when PE is conceived as the development of sport techniques [60] or to develop elite sporting performance [3]. These may not be aligned with the inclusive and socially just approaches of PL and the capabilities approach whereby all children and young people should be nurtured to develop and realise their individual potentials, with PE for all, and not the preserve of a few.

PE has the potential to significantly contribute to the development of all ten capabilities when enriched by physical literacy. However, one factor that may have hindered the progress of PE in school curricula could be through the justification of its value in serving as a means to achieve other (quite numerous and sometimes superficial) ends. These may include that it provides a break from ‘academic’ work, it provides essential physical activity for pupils, it can tackle the inactivity pandemic or obesity pandemic, and it gives potential exam success for less academic pupils. This series of instrumental ends are far off the mark of a capabilities approach to human dignity or social justice [16]. The capabilities approach does set challenges for systemic structural issues in many societies, but these are merely introduced here to prompt further discussion. In the context of PE curricula, however, we heed the warning by Tinning [61] among others that we should be more modest in the educational claims made on behalf of PE. Bailey et al. [62] argue that educational claims of PE should be clear, being mindful of the responsibility for delivering a multitude of outcomes. In essence, rather than proposing that all ten capabilities should inform PE curricula and pedagogy, we suggest that awareness of different ways to think about PL-enriched PE could provide useful support for enhancing social justice at a very specific level. Those using PL as the foundation of PE might also reflect on the capabilities approach as a way of widening engagement and participation by opening up conceptions of PE rather than closing them down.

## 15. Conclusions

In this paper, we have considered the relationships between the capability approach, PL, and PE, and included a posthuman perspective. Robeyns [11] (p. 10) proposes that the capability approach can be used to evaluate “the lives of individuals and the societies in which these people live their lives”. In ‘evaluation’, a PL-enriched approach to PE may indeed provide further and multiple opportunities for capabilities to be developed, but rarely do educators consider the capabilities approach when curriculum planning, delivering lessons, or evaluating their PE offer. Instead, if the correlation between nurturing PL and the development of the capability approach can be considered connected, then it reinforces the value of physical educators using PL as a valid framework for promoting lifelong engagement in physical activity for health and flourishing.

This paper has positioned the value of PL as a fulcrum concept that can help to foster capabilities and enhance PE in pursuit of lifelong engagement in physical activity for health and flourishing. Sen [63] (p. 54) states “that a capability is the ability or potential to do or be something, more technically, to achieve a certain functioning”. PL can be considered a capability because it aims to capitalise on our movement potential and embodied capability, without which a flourishing and fulfilled life may not be possible. Rationalising the relationship between nurturing PL and the simultaneous development of capabilities is a powerful conceptualization, especially in elevating the value of PE when enriched by PL. Not only does the capability approach offer value to PL and PE, but so do PE and PL feed into the value of the capability approach as a meaningful construct with which to explore and understand the posthuman world.

While the informal link between PL and capabilities has been made previously [4,31], this paper is significant because it is the first of its kind to explore this potential relationship in more detail. This paper recommends that the capabilities approach offers a valuable framework for the continued justification of PL-enriched PE that, when aligned, can help to shape the opportunities provided for children and young people in support of their holistic development and lifelong engagement in physical activity.

## Figures and Tables

**Figure 1 children-10-01503-f001:**
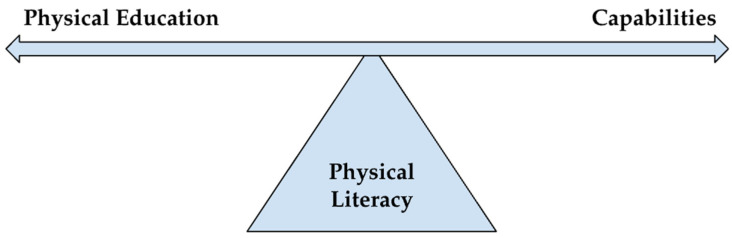
Physical literacy as the fulcrum for PL-PE and capabilities.

## Data Availability

Not applicable.

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
