# Peer review of "Physical-Literacy-Enriched Physical Education: A Capabilities Perspective"

_children, 2023, doi:10.3390/children10091503_

Round 1

Reviewer 1 Report

Thank you for giving me the chance to review the paper ‘Expanding the Philosophical Foundations of Physical Literacy: Considerations for Developing a Moral Sense of Self’. I would like to be transparent in admitting I would not consider myself to be an expert regarding the capability approach, but I find discussions regarding the philosophy of physical literacy valuable and meaningful. This is often an area that gets criticised for being ‘inaccessible’, so as an enthusiastic non-expert I have made some suggestions, but acknowledge that narrative should definitely not be simplified if this loses or changes meaning.

I enjoyed reading this paper, and in particular section 4 onwards. While I understand the need for the detailed opening for context, I did find myself having to re-read sections 2 and 3, and suggest these could be edited. I’m reluctant to just say condensed here, but this is quite a long paper, and I do think the major contributions come in the later parts of the article. There are also points where I found the longer quotes disrupted the author's narrative (e.g page 4 line 204-209).

The discussion and conclusion are well written to round off detailed and complex narrative. Line 613-631 for example nicely brings the discussion back to the practical implications/debates regarding PE.

Please see attached for minor line by line comments. I hope you find my feedback helpful.

Author Response

Thank you for reviewing our paper. Please see our responses in the table attached. 

Reviewer 2 Report

Dear authors, 

Having reviewed the research, I consider it to be evidence of high novelty. However, I have a number of questions. 

If it is a qualitative study, why does the research lack material and method?

As I have seen, I would consider it more interesting to carry out a systematic review to better understand the problems presented. 

I have a lot of doubts, as this research lacks methodology. 

If the research does not clearly explain how the data analysis was carried out, it cannot be publishable. 

Author Response

Thank you for reviewing our paper. Please see our response below with regard to your concerns over the lack of a method.

This work is post-qualitative, not qualitative, which means that a traditional methodological approach is challenged in favour of thinking with theory. This approach is addressed in the positioning section and is not uncommon in this type of theoretical work. We have included references to this approach (19-26).

  • Pierre, E. A. (2019) Post qualitative inquiry in an ontology of immanence. Qualitative Inquiry, Vol. 25, No. 1, 3-16, DOI: 10.1177/1077800418772634
  • Pierre, E. A. (2013) The posts continue: becoming. International Journal of Qualitative Studies in Education, Vol. 26, No. 6, 646-657, DOI: 10.1080/09518398.2013.78875
  • Kuntz, A. M. (2020) Foucauldian practices: Philosophical inquiry as virtuous enactments for material change. Access: Contemporary Issues in Education. Vol. 40, No. 1, 41–46. DOI: 10.46786/ac20.8961
  • Taguchi, H., L. and St. Pierre, E. (2017) Using Concept as Method in Educational and Social Science Inquiry. Qualitative Enquiry, Vol. 23, No. 9, 643-648. DOI: 10.1177/1077800417732634
  • Collins, C. S. and Stockton, C. M. (2018) The Central Role of Theory in Qualitative Research. International Journal of Qualitative Methods, 17(1), pp. 1–10. DOI: 10.1177/1609406918797475 
  • Bignall, S. (2010) Affective Assemblages: Ethics beyond Enjoyment. In Bignall, S. and Patton, P. (eds.). Deleuze and the Postcolonial, Chapter 4, pp. 78-102. Edinburgh University Press. ProQuest Ebook Central. Retrieved from:  http://ebookcentral.proquest.com/lib/stir/detail.action?docID=564500.
  • Barad, K. (2007) Meeting the Universe Halfway. Quantum Physics and the Entanglement of Matter and Meaning. Durham-London: Duke University Press.
  • Deleuze, G. and Guattari, F. (1987) A Thousand Plateaus. Trans. B. Massumi, Minneapolis: Minnesota University Press

Reviewer 3 Report

The results of the study are essential. The authors have discussed the issue well. The originality of the work is high. Investigation of decision-making styles of individual and team sports coaches.

Author Response

(The authors gave the same response as above.)

Round 2

Reviewer 2 Report

Dear Authors, 

Thank you very much for your response. I understand your point of view, but a scientific research has to present a consistent methodological framework. In this case this type of research does not offer such a framework. It could be reconsidered if it focused on a systematic review study through which the proposed subject matter is exposed. 

Author Response

Please refer to this article published in the same journal - post-qualitative research is a different research paradigm than positivist - systematic reviews. Both offer contributions to knowledge but from different perspectives.

https://www.mdpi.com/2227-9067/10/3/497